# Diagnostic Approach to Pulmonary B-Cell Lymphomas in Small Biopsies, with Practical Recommendations to Avoid Misinterpretation

**DOI:** 10.3390/diagnostics13213321

**Published:** 2023-10-26

**Authors:** Sergio Pina-Oviedo, Victor L. Roggli, Thomas A. Sporn, Huihua Li, Carolyn Glass, Louis R. DiBernardo, Elizabeth N. Pavlisko

**Affiliations:** Department of Pathology, Duke University Medical Center, Durham, NC 27710-4000, USAelizabeth.pavlisko@duke.edu (E.N.P.)

**Keywords:** MALT lymphoma, DLBCL, lymphomatoid granulomatosis, intravascular lymphoma, differential diagnosis, immunohistochemistry, extranodal lymphoma, nodular lymphoid hyperplasia

## Abstract

Pulmonary lymphomas are rare. With the current less invasive approaches used to obtain material for diagnosis, the diagnosis of pulmonary lymphoma is now frequently established in a small biopsy rather than in a resection. Therefore, the diagnosis has become more challenging and requires correlation with the clinico-radiologic presentation and with ancillary studies (immunohistochemistry, flow cytometry, cytogenetics, and/or molecular analysis). Due to the rarity of pulmonary lymphomas, clinical suspicion of a lymphomatous process is low at initial presentation, and material may be only submitted for histopathology. For this reason, herein, we provide recommendations to arrive at the correct diagnosis of the most common lung B-cell lymphomas (marginal zone lymphoma of mucosa-associated lymphoid tissue, diffuse large B-cell lymphoma, intravascular large B-cell lymphoma, lymphomatoid granulomatosis) in the setting of small biopsies, utilizing only immunohistochemistry. The differential diagnosis varies according to the lymphoma subtype and includes reactive conditions, solid tumors, and other hematolymphoid malignancies. Although morphology and immunohistochemistry may be sufficient to establish a diagnosis, in some cases, the best recommendation is to obtain additional tissue via a VATS biopsy/wedge resection with material submitted for flow cytometry, cytogenetics, and/or molecular studies to be able to properly classify a pulmonary lymphoid process.

## 1. Introduction

Pulmonary lymphomas are rare. They can be divided into primary disease, when the lymphoma is confined to the lung and to the hilar lymph nodes, or as secondary disease, when lung involvement is part of systemic disease. Secondary lung involvement is by far more common than primary disease (<1% of primary lung neoplasms). Pulmonary lymphomas are more common in adults than in children, with a median age of 60 years [1,2].

Given the current less invasive approaches used to obtain material for the diagnosis of lung tumors, the diagnosis of pulmonary lymphoma is now more frequently established in a transbronchial biopsy, a core needle biopsy, or a fine-needle aspiration rather than in a surgical resection. Because the amount of tissue in these kinds of specimens is smaller than in a resection, the diagnosis becomes much more challenging and usually requires close correlation with the clinico-radiologic presentation (prior history of lymphoma or not; interstitial pattern with ground-glass opacities or reticulations versus one or multiple masses on imaging; presence or absence of regional lymphadenopathy) and with ancillary studies (immunohistochemistry (IHC), flow cytometry, cytogenetics, and/or molecular analysis). However, due to the rarity of pulmonary lymphomas, there is usually a low clinical suspicion for a lymphomatous process involving the lungs at initial presentation, and material may be only submitted for histopathology. This is also true for ground-glass lesions measuring <3 cm that may be resected and submitted for permanent sections with a presumptive diagnosis of lepidic adenocarcinoma.

With the above information in mind, herein, we present a diagnostic approach to some of the most frequent lung B-cell lymphomas (marginal zone lymphoma of the mucosa-associated lymphoid tissue, diffuse large B-cell lymphoma, intravascular large B-cell lymphoma, and lymphomatoid granulomatosis) as compared to some of their reactive or other malignant mimickers (nodular lymphoid hyperplasia, lymphoid interstitial pneumonia (LIP), primary or metastatic lung cancer, lymphangitic carcinoma, vasculitis, necrotizing infection, and necrotizing sarcoid) based predominantly on the refined use of morphology and immunohistochemistry when assessing a small biopsy (summarized in Table 1).

Similarly, we give recommendations as to when cytogenetics and/or molecular studies may be helpful for diagnosis. Last, in some instances, it may not be possible to establish a diagnosis of lymphoma on a small biopsy, and the best decision from a pathologist will be to recommend an excisional procedure/wedge resection via video assisted thoracoscopy (VATS) with emphasis on collecting fresh material to perform flow cytometry, cytogenetics, and/or molecular analysis. A retrospective study from Italy showed that a transbronchial cryobiopsy may also be an effective tool in the diagnosis of pulmonary lymphomas [3]. Plasma cell neoplasms and post-transplant lymphoproliferative disorders are not discussed in this review.

## 2. Marginal Zone Lymphoma of Mucosa-Associated Lymphoid Tissue (MALT Lymphoma)

MALT lymphoma is the most common type of primary pulmonary lymphoma (70–80% of cases) [1,2,4]. The disease primarily affects adults (median age 68 years) [5], but it may develop in younger individuals who are immunosuppressed or are infected with the human immunodeficiency virus. About a third of patients have an underlying autoimmune disorder, such as Sjögren syndrome, rheumatoid arthritis, or lupus erythematosus, and a similar proportion are asymptomatic. The association between the Gram-negative bacterium *Achromobacter xylosoxidans* and pulmonary MALT lymphoma is controversial [6,7]. Symptomatic individuals have non-specific findings that are confused with pneumonia, such as fever, cough, dyspnea, or hemoptysis, but these symptoms do not respond to antibiotics.

On a chest X-ray, pulmonary MALT lymphoma usually presents as a solitary lung mass. On high resolution computed tomography (CT) scan and magnetic resonance imaging (MRI), this lymphoma may present as a ground glass infiltrate with air bronchograms (Figure 1), as a peripheral mass with pleural thickening [8,9,10], or as consolidated lung parenchyma with air bronchograms and areas of apparent cavitation [11,12,13].

Multiple or bilateral lesions, a pattern resembling LIP, or endobronchial lesions may occur but are uncommon [10,14]. Hilar or mediastinal lymphadenopathy is seen in ~30% of cases [15].

*Histopathologic features in biopsies.* MALT lymphoma is a polymorphic lymphoma composed predominantly of sheets of small lymphocytes, centrocyte-like cells, and monocytoid cells, with few scattered large immunoblast-like cells and a variable number of plasma cells. This process typically effaces the lung parenchyma (Figure 2 and Figure 3).

Residual distorted lymphoid follicles “colonized” by monocytoid cells or reactive lymphoid follicles with prominent marginal zones may be seen. Lymphoepithelial lesions involving the airways may or may not be appreciated depending on the extent of involvement and if an airway was sampled in the biopsy (Figure 3).

However, lymphoepithelial lesions are not exclusive of lymphoma and may be seen in reactive conditions. Plasma cells may be located beneath the airway epithelium or may form small clusters admixed with lymphocytes. A few multinucleated giant cells, non-necrotizing granulomas, or areas of fibrosis may be present.

*Differential diagnosis.* If the lesion is mass-forming on imaging, the polymorphic nature of the specimen and the presence of plasma cells should raise concern for MALT lymphoma versus a reactive and dense lymphoplasmacytic process, such as non-specific chronic inflammation, nodular lymphoid hyperplasia, or IgG4-related lung disease (IgG4-RLD) [16,17]. The required IHC in this scenario includes CD3, CD20, CD10, bcl-2, bcl-6, CD21/CD23, IgG, IgG4, and kappa and lambda by IHC or in situ hybridization (ISH).

In a reactive process, including nodular lymphoid hyperplasia and IgG4-RLD, there is a predominance of CD3 over CD20 with no CD20+ intraepithelial lymphocytes, and the B-cells are negative for bcl-2. CD10 and bcl-6 highlight germinal centers that are negative for bcl-2. CD21/CD23 are positive in normal follicular dendritic cell meshworks, and plasma cells are always polytypic by IHC and/or ISH. Pan-cytokeratin shows no lymphoepithelial lesions (Figure 4).

In a biopsy, the lymphoplasmacytic infiltrate of LIP or severe lung involvement by a connective tissue disease may be quite brisk to raise concern for MALT lymphoma (Figure 5).

Lack of lymphoepithelial lesions or numerous B-cells and the presence of polytypic plasma cells in the setting of ground-glass opacities instead of a mass favor LIP or lung involvement by connective tissue disease over MALT lymphoma (Figure 5). As in all extranodal plasma cell-rich polytypic processes, it is important to exclude IgG4-RLD by evaluating the IgG4:IgG ratio. The criteria to suggest the possibility of IgG4-RLD consists of >20 IgG4+ plasma cells/high power field and an IgG4:IgG ratio > 40% [18]. However, in IgG4-RLD, there are usually areas of fibrosis and scattered eosinophils with/without the presence of vasculitis. The latter is not a feature typically seen in MALT lymphoma or nodular lymphoid hyperplasia. Importantly, a subset of cases of nodular lymphoid hyperplasia and MALT lymphoma can also have increased IgG4+ plasma cells outside of the context of IgG4-RLD. Therefore, pathologists should be careful about establishing a diagnosis of IgG4+-RLD without a proper morphologic evaluation and correlation with clinical and radiologic findings. For cases with features of nodular lymphoid hyperplasia and increased IgG4+ plasma cells, it is better to give a descriptive diagnosis of “lymphoplasmacytic infiltrate with increased IgG4+ plasma cells” and recommend clinical and laboratory correlation before establishing a diagnosis of IgG4-RLD [19]. Recommended laboratory studies include serology for IgG subclasses, IgE, and a thorough work up to exclude an underlying autoimmune disorder. See below for cases of MALT lymphoma with increased IgG4+ plasma cells.

In MALT lymphoma, contrary to a reactive process, there is usually CD20 predominance over CD3, although not always (Figure 6).

The neoplastic B-cells co-express bcl-2 and are negative for CD10 and bcl-6. Distorted germinal centers may show residual labeling with CD10 and bcl-6, with a variable number of bcl-2+ lymphoma cells colonizing follicles (Figure 6). CD20 or pan-cytokeratin highlight lymphoepithelial lesions if present (Figure 6). Plasma cells may or may not be monotypic; therefore, it is crucial to know that a lack of monotypic plasma cells does not rule out a diagnosis of MALT lymphoma (Figure 6). Similarly, it is not uncommon to have increased IgG4+ plasma cells in MALT lymphomas, which may be misinterpreted as IgG4-RLD [20]. This is readily excluded when the IgG4+ plasma cells are monotypic. If a case with increased IgG4+ plasma cells is polytypic, using the other features described above should be sufficient to support a diagnosis of MALT lymphoma over IgG4-RLD. However, if no convincing morphologic or immunophenotypic features supportive of MALT lymphoma are seen, the diagnosis may be rendered as “lymphoplasmacytic infiltrate, cannot exclude MALT lymphoma”. Despite the traditional recommendation to perform *IGH* clonality studies in these kinds of cases to favor a reactive or malignant process, it should be clarified that these results should never be used to make a definitive diagnosis of benign or malignant. Alternatively, fluorescence in situ hybridization (FISH) looking for the t(11;18)(q21;q21)(*API2::MALT1*) may be helpful for diagnosis since this is the most common genetic alteration in lung MALT lymphoma; however, this translocation is only present in 40–50% of cases (only helpful if it is detected) [21]. Close follow up of these patients and an attempt for a new biopsy with material submitted for flow cytometry may be the best recommendation.

Some MALT lymphomas composed of a more monotonous lymphocytic infiltrate are hard to distinguish morphologically from other low-grade B-cell lymphomas, including follicular lymphoma, chronic lymphocytic leukemia/small lymphocytic lymphoma, and mantle cell lymphoma. Usually, these lymphomas tend to involve the lungs at advanced stages, and there is already a known diagnosis. However, rarely, lung involvement may be the initial clinical manifestation, and thus, a biopsy may be performed. The required IHC in this scenario includes CD3, CD20, CD5, CD10, bcl-2, bcl-6, CD21/CD23, CD43, cyclin D1, and Ki-67. MALT lymphoma is a CD5−/CD10− B-cell lymphoma with co-expression of bcl-2 in B-cells. About 40–50% of cases may also co-express CD43 (Figure 6).

MALT lymphoma with numerous “colonized” follicles may be difficult to distinguish morphologically from low-grade follicular lymphoma. The presence of a polymorphic tumor composed of monocytoid cells, plasma cells, and lymphoepithelial lesions supports MALT lymphoma over follicular lymphoma, whereas in follicular lymphoma the tumor is composed of variable number of centrocytes and centroblasts. IHC helps to distinguish this neoplasm since follicular lymphoma is positive for CD10 and bcl-2 and negative for CD5, CD23, CD43, and cyclin D1. However, prominent follicular colonization with numerous bcl-2+ lymphoma MALT cells may not be readily distinguished from the bcl-2+ neoplastic follicles of follicular lymphoma, and CD10 and bcl-6 may be difficult to interpret as either residual germinal centers or as lymphoma cells. Additionally, some cases of follicular lymphoma can show marginal zone differentiation and are negative for CD10 and bcl-6. In these difficult cases, it is recommended to perform FISH to detect the characteristic translocation of follicular lymphoma t(14;18)(*BCL2::IGH)* to attempt to confirm the diagnosis, but this translocation may also be absent in follicular lymphomas with marginal zone differentiation. Alternatively, FISH looking for the t(11;18)(q21;q21)(*API2::MALT1*) may be helpful for diagnosis since this is the most common genetic alteration in lung MALT lymphoma; however, this translocation is only present in 40–50% of cases [21]. If it is not possible to discern between these two lymphomas, a diagnosis of “CD5−/CD10− low-grade B-cell lymphoma” may be sufficient. If a more refined diagnosis is needed and no other sites of involvement are detected, molecular studies may prove helpful to identify alterations seen more frequently in lung MALT lymphoma (*TBL1XR1* and *TET2* mutations) than in follicular lymphoma (*KMT2D*, *EZH*, *CREBBP,* and *EP300* mutations) [22].

Chronic lymphocytic leukemia/small lymphocytic lymphoma is composed of small monotonous lymphocytes with clumped chromatin and proliferation centers containing prolymphocytes and paraimmunoblasts that may appear as nodular pale areas within the tumor, mimicking MALT lymphoma morphologically. However, the lymphoma B-cells are positive for CD5, and CD23 but lack CD10, CD21, and cyclin D1, and this confirms the diagnosis and excludes MALT lymphoma (Figure 7).

Mantle cell lymphoma is a B-cell lymphoma composed of small lymphocytes with indented nuclei that may resemble centrocyte-like cells. In addition, this lymphoma may show a vague nodular arrangement, hyalinized venules, and epithelioid macrophages. By IHC, mantle cell lymphoma is positive for CD5 and cyclin D1 and is negative for CD10 and CD23. This immunophenotype rules out MALT lymphoma.

Lymphoplasmacytic lymphoma also enters the differential diagnosis of MALT lymphoma since this is also a CD5−/CD10− B-cell lymphoma, but in the lung, this neoplasm is extremely rare, and it should only be considered as a diagnosis of exclusion or if the patient had a prior diagnosis of lymphoplasmacytic lymphoma.

Cases of MALT lymphoma with extensive plasmacytic differentiation enter the differential diagnosis of lung plasmacytoma (solitary plasmacytoma or extramedullary plasma cell myeloma). The presence of small lymphocytes and monocytoid cells (both CD20+), reactive follicles, and lymphoepithelial lesions is diagnostic of MALT lymphoma and excludes plasmacytoma, where only sheets of plasma cells are seen. However, in a core biopsy, not all these features may be present [23,24,25]. By IHC, a plasma cell neoplasm (either solitary plasmacytoma or extramedullary plasma cell myeloma) is favored over MALT lymphoma if the plasma cells are positive for CD56 and CD117 and lack CD45. Amyloid deposition may be observed in both entities and can be confirmed by Congo red stain.

If increased large cells are seen in MALT lymphoma, but criteria for diffuse large B-cell lymphoma are not met (sheets of large cells), this should be documented and discussed with the clinical team since these cases behave more aggressively than classic MALT lymphoma.

Importantly, the edges of a pulmonary MALT lymphoma are ill-defined and show alveolar septal expansion with numerous lymphocytes with variable degrees of alveolar edema, intra-alveolar proteinaceous debris, and reactive pneumocyte changes (Figure 8).

These features are morphologically indistinguishable from LIP (Figure 8). Therefore, mandatory correlation with imaging is needed in a transbronchial or core needle biopsy with morphologic features of LIP where the imaging shows a mass rather than diffuse ground-glass opacities. In this setting, CD3 and CD20 IHC should be performed to document the type of predominant lymphocytes (LIP usually has more CD3+ than CD20+ cells, whereas MALT lymphoma is the opposite). We recommend signing cases of nodules or masses with morphologic features of LIP in a biopsy as ”prominent septal alveolar lymphocytic infiltrate, B-cell-rich or T-cell-rich” (based on the IHC results) and comment that the possibility of MALT lymphoma cannot be entirely excluded, particularly if the lymphoid infiltrate is B-cell-rich. Close clinical and imaging correlation as well as a recommendation for additional material with a sample submitted for flow cytometry are encouraged. Lastly, the edges of nodular lymphoid hyperplasia are typically sharply demarcated from the lung (Figure 8) in contrast to the findings described for LIP and MALT lymphoma, but this is difficult to appreciate in a core biopsy.

## 3. Diffuse Large B-Cell Lymphoma (DLBCL)

DLBCL is the second most common pulmonary lymphoma (10–20% of cases) [2,4]. Most cases of primary lung DLBCL correspond to large cell transformation of pulmonary MALT lymphoma, whereas the pathogenesis of de novo DLBCL is unclear. DLBCL is a disease of adults, with a mean age of 60 years. Contrary to MALT lymphoma, most patients present with B symptoms. On imaging, primary pulmonary DLBCL tends to present as a single peripheral mass with sharply demarcated borders, but this may not be the case for a DLBCL arising from MALT lymphoma that may present as an ill-defined mass. DLBCL may present as multiple lung nodules, usually in the setting of systemic disease. On CT scan, DLBCL can show central necrosis [15], and about 50% of patients have hilar lymphadenopathy.

*Histopathologic features in biopsies.* DLBCL is composed of sheets of large lymphoid cells with centroblastic and immunoblastic features (Figure 9).

Centroblastic morphology refers to cells with an oval to irregular nucleus with fine chromatin, >1 basophilic nucleoli with a juxtanuclear membrane location, and a moderate amount of amphophilic to pale eosinophilic cytoplasm. Immunoblastic features refer to cells with a round nucleus with fine chromatin, a central eosinophilic nucleolus, and radial projections of “wispy” chromatin threads and more abundant cytoplasm, sometimes with plasmacytoid morphology (Figure 9). The tumor produces effacement of the lung architecture, and there are abundant mitoses and apoptotic bodies and variable degrees of necrosis. In areas with preserved lung parenchyma, the tumor cells may be seen admixed with fibrin and filling the alveolar spaces, a phenomenon called “tumoral pneumonia” (Figure 9) [26].

*Differential diagnosis.* The morphology of the tumor is highly suggestive of large cell lymphoma, and a limited panel of IHC markers may solve the issue, including CD45, CD3, and CD20, that will point to a diagnosis of DLBCL (Figure 10).

However, in some instances, the morphologic features may not be obvious or may be obscured by crush artefact and/or necrosis. The presence of anaplastic cells or a “starry sky” pattern may raise concern for a primary lung or metastatic carcinoma, melanoma, small cell carcinoma (Figure 11), or small blue round cell tumors metastatic to the lung, including but not limited to Ewing sarcoma, neuroblastoma, embryonal rhabdomyosarcoma, etc.

Curiously, in biopsies, DLBCL tends to show spindle morphology mimicking sarcomatoid carcinoma or sarcoma. In these cases, IHC for pan-cytokeratin, S100, and CD45 is required to first classify a case as a hematolymphoid neoplasm and exclude carcinoma or melanoma. Other specific markers are required if a hematolymphoid origin is excluded, e.g., FLI1, CD99, and NKX2.2 for Ewing sarcoma; neuroendocrine markers for neuroblastoma; desmin and myogenin for embryonal rhabdomyosarcoma; etc.

Once confirmation of a hematolymphoid origin is established with CD45, the ideal IHC lymphoma work up for DLBCL includes CD3, CD20, CD5, CD10, bcl-2, bcl-6, MUM1, cyclin D1, CD30, Ki-67, and Epstein-Barr virus-encoded RNA (EBER) ISH. These markers are the minimum to best classify a lymphoma as of B-phenotype (CD20+,CD3−), from germinal center origin (CD10+, bcl-6+, MUM1-) or non-germinal center origin (CD10−, bcl-6−/+, MUM1+), as a “double expressor” or not (bcl-2+ and c-myc+), as an EBV+ LBCL, and to exclude the diagnosis of blastoid or pleomorphic mantle cell lymphoma (CD5+ and cyclin D1+). Knowing the cell of origin of a DLBCL (germinal center vs non-germinal center) as well as the status of bcl-2 and c-myc (“double expressor”) confers prognostic value, although this has not been properly studied specifically in primary pulmonary DLBCL. The expression of CD30 can be used to record the potential use of targeted treatment with the anti-CD30 antibody therapy, brentuximab vedotin. Ki-67 is helpful to know the proliferation index. If Ki-67 is >90%, this may suggest a diagnosis of high-grade LBCL. If not all the above markers can be performed, we recommend 1) attempting to determine the B-cell lineage of the tumor by performing CD20 or PAX5 and 2) by performing cyclin D1 IHC to at least exclude mantle cell lymphoma. PAX5, CD20, and CD79a highlight the intra-alveolar malignant cells in areas of “tumoral pneumonia” (Figure 10). Careful interpretation of PAX5 without confirmation of a hematolymphoid origin (CD45, CD43) should be kept in mind since up to 30% of small cell carcinomas can be positive for PAX5 [27], and some cases of small cell carcinoma may be negative or only patchy positive for keratins and synaptophysin.

Rarely, myeloid sarcoma may present as a lung mass mimicking DLBCL on morphology. Clinical history of prior or concurrent acute myeloid leukemia is extremely helpful to consider this possibility. IHC for CD34, CD117, MPO, and myeloid or monocytic markers (CD33, CD14, CD4) as well as for B-cell markers (CD20, CD79a, PAX5) is helpful to confirm the diagnosis.

A more challenging differential diagnosis of pulmonary DLBCL is with other large cell lymphomas involving the lung, such as grade 3 lymphomatoid granulomatosis (LyG), primary mediastinal (thymic) LBCL extending into the lung, Burkitt lymphoma, or anaplastic large cell lymphoma. Grade 3 LyG shows an angiocentric pattern that is not typical of DLBCL, but this may not be present in a core needle or transbronchial biopsy. Grade 3 LyG is strongly and diffusely positive for EBER, which is negative in DLBCL unless the case is an EBV+ LBCL, and that distinction requires clinical and imaging correlation (see section “Differential diagnosis of high-grade LyG”).

Primary mediastinal (thymic) LBCL extending to the lung should be ruled out by clinical and imaging correlation, such as the case of a young woman with an anterior mediastinal mass that is extending into the lung, versus that of an older man or an adult with a single peripheral lung nodule or multiple lung nodules [28]. This lymphoma shows clear cells and variable degrees of fibrosis, and these features may or may not be observed in cases involving the lung (Figure 12).

By IHC, primary mediastinal (thymic) LBCL is positive for B-cell markers, and for MUM1, CD23 and p63 in about 70% of cases with variable to weak expression of CD30.

Burkitt lymphoma is a CD10+ B-cell lymphoma of intermediate-sized cells that also enters the differential diagnosis of a DLBCL of germinal center origin (CD10+). This diagnosis should be considered when a LBCL is negative for bcl-2 and shows strong expression of c-myc with a “starry sky” pattern and a Ki-67 close to 100%. EBV may be positive or negative. Confirmation or exclusion of the diagnosis requires correlation with FISH and/or molecular analysis. This discussion is out of the scope of this review, but it is worth mentioning that pulmonary involvement by Burkitt lymphoma is exceedingly rare as compared to that of DLBCL.

For anaplastic large cell lymphoma, the use of CD3 and other T-cell markers, as well as CD4, CD8, CD20, PAX5, ALK, and CD30, are helpful for diagnosis. Anaplastic large cell lymphoma is a CD30+ T-cell lymphoma, usually positive for CD2, CD4, and variable expression of CD5, CD7, and CD8 but negative for B-cell markers. About half of the cases are ALK+. It is important to be aware that the anaplastic variant of DLBCL is also positive for CD30, but the tumor expresses B-cell (not T-cell) markers.

## 4. Intravascular Large B-Cell Lymphoma (IV-LBCL)

Intravascular LBCL is a rare subtype of LBCL with specific clinicopathologic features. This neoplasm shows characteristic intravascular involvement by large lymphoma cells in multiple organs, hence the name. A lack of homing receptor molecules, such as CD29 (beta-1 integrin) and CD54 (ICAM-1), on the surface of lymphoma cells impairs their ability to exit the tissues and partially explains their peculiar intravascular location [29]. The most common affected sites include the skin, the lungs, and the central nervous system, but any organ may be involved by IV-LBCL. Lung involvement is part of the so-called “Asian form of IV-LBCL” that features multisystemic involvement along with hemophagocytic syndrome [30,31,32]. The disease shows non-specific clinical and imaging findings, and only a high index of suspicion will point toward this diagnosis, which may only happen after several other conditions have been excluded. Unfortunately, most patients go underrecognized and the histopathologic diagnosis is established at late stages of the disease or only at autopsy. Overall, IV-LBCL has a poor prognosis despite the use of current DLBCL-based chemotherapy.

*Histopathologic features in biopsies.* IV-LBCL may be inconspicuous with only a few lymphoma cells seen within blood vessels, or it may show a patchy distribution (Figure 13).

For these reasons, the diagnosis can be easily overlooked, and only a high index of suspicion will point to this diagnosis. On the contrary, cases with high tumor burden produce expansion of blood vessels that appear as alveolar septal expansion and may be interpreted as an interstitial pneumonic process [33]. The lymphoma cells show similar morphologic features to those described for DLBCL (centroblastic or immunoblastic) and are seen in the lumen of capillaries and of small- to intermediate-sized blood vessels. In some cases, the lymphoma cells are seen floating in the lumen, whereas in other cases, they may “pack” the entire blood vessel lumen, show “margination”, or may be admixed with fibrin or a thrombus [4]. IV-LBCL is positive for CD45, CD20, PAX5, CD79a, and bcl-2, with variable expression of CD5, CD10, bcl-6, and MUM1. CD15 and CD30 are usually negative (Figure 13). Most cases have a non-germinal center type immunophenotype, CD10−/bcl-6+/−/MUM1+. The lymphoma cells are negative for T-cell markers, ALK, EBER, keratins, and melanoma markers.

*Differential diagnosis.* IV-LBCL should be distinguished from carcinoma or melanoma with lymphangitic spread (Figure 14), acute leukemia (usually cases with hyperleukocytosis), and an intravascular large T-cell lymphoma (usually NK/T-cell lymphoma or anaplastic large cell lymphoma) [34].

A proper set of IHC is sufficient to make the correct diagnosis, which includes CD45, ≥1 B-cell markers (CD20, PAX5, CD79a), CD3, CD56, EBER, pan-cytokeratin, or S100 (Figure 14). If the consideration of leukemia is high, the addition of CD34, CD117, MPO, or other myeloid or monocytic markers (CD33, CD14, CD4) may be helpful. We recommend to always consider IV-LBCL as a potential diagnosis in cases where there appears to be no significant pathologic findings or there only appears to be mild alveolar septal expansion in the lung biopsy and the patient has severe systemic symptoms (weight loss, shortness of breath), and an unimpressive imaging or one with ground-glass opacities. The utility of CD20 in this context is highly valuable and confirms the diagnosis.

## 5. Lymphomatoid Granulomatosis (LyG)

LyG is a clonal proliferation of EBV-infected large B-cells associated with vasculitis and necrosis [35,36]. LyG predominantly affects males (M:F ratio 2:1), with a mean age of presentation of 46 to 48 years, who are immunosuppressed, including patients with acquired immunodeficient syndrome, who are status post ablation chemotherapy, and patients with hereditary immune deficiencies (e.g., Wiskott–Aldrich syndrome) [37,38,39,40]. Rarely, immunocompetent individuals may also develop LyG. This disease affects the lungs commonly (~85% of cases), followed by involvement of the brain, skin, and kidneys, but with peculiar sparing of the reticuloendothelial system [35]. Clinical symptoms include cough, dyspnea, chest discomfort, hemoptysis, and fever, and these symptoms wax and wane for months to years [35,36]. On imaging, most patients show bilateral lung nodules ranging from 0.5 cm to >10 cm, with a preferential peribronchial and vascular distribution. The nodules become confluent over time and develop central cavitation and necrosis [8,15]. Over time, well-developed nodules may decrease in size or show “regression”, leaving only a ground glass opacity indicative of a previous focus of active disease, while new nodules may appear simultaneously. This is the reason why they have been referred to as “migratory” nodules [15]. Rarely, LyG may present as a single lesion. Hilar lymphadenopathy is not seen.

*Histopathologic features in biopsies.* The diagnosis of LyG in core biopsies is challenging. LyG is classified on histopathology into 3 grades: grades 1 and 2 are considered “low-grade”, whereas grade 3 is considered a “high-grade” lesion. They all vary on the amount of large, atypical EBER+ B-cells present and the amount of necrosis. Due to the presence of necrosis, transbronchial biopsies are diagnostic for LyG only in ~30% of cases.

Grades 1 and 2 LyG vary from lesions composed of a polymorphic infiltrate of small lymphocytes, plasma cells, and macrophages, with a few scattered large atypical lymphoid cells with immunoblastic or Reed–Sternberg-like features (Figure 15).

Grade 1 cases have <5 large atypical EBV+ B-cells/10 high power fields with no necrosis, whereas grade 2 lesions contain 5–20 EBV+ B-cells/10 high power fields and focal necrosis (Figure 15). However, in a core biopsy, these features may be difficult to appreciate, and lesions may just be reported as “grade 1–2/low-grade LyG”. On the other hand, grade 3 LyG is composed of sheets of large EBV+ B-cells with extensive necrosis (Figure 16 and Figure 17).

Additionally, a common denominator in LyG is the presence of lymphocytic vasculitis. In cases with necrosis, the latter is of coagulative/eosinophilic type and lacks neutrophils or karyorrhectic debris. Despite its name, LyG is devoid of granulomas (the term “granulomatosis” is a misnomer), and eosinophils and neutrophils are infrequent. If the edge of a LyG nodule is sampled, there is usually a sharp demarcation from the adjacent normal lung parenchyma. In a core biopsy, the diagnostic accuracy depends on the identification of large EBER+ B-cells. In a case with clinical and radiologic features of LyG where the right background is seen but no large EBER+ B-cells are noted, it is recommended to perform additional deeper levels to try to find these cells. If a “regressed” nodule is biopsied, the findings are non-specific, showing fibrosis with recanalized blood vessels, scant chronic inflammation, and adjacent hemorrhage and lung edema, with no residual large cells [35].

By IHC, the large B-cells are positive for CD20, PAX5, and CD79a, and for EBER by ISH and LMP1 by IHC, indicating a type II latency of EBV infection (Figure 16).

The large atypical cells are negative for T-cell markers and CD15, with variable expression of CD30 [37]. The background infiltrate is composed of CD3+ small T-cells (CD4 > CD8). CD68 highlights macrophages and CD138, and kappa and lambda highlight polytypic plasma cells. Special stains are negative for microorganisms.

Grading LyG using EBER is controversial since cases with extensive necrosis may be falsely negative for detection of the virus RNA. Likewise, there have been reports of EBV-negative LyG cases, which pose a problem using EBER+ cells for grading [38]. Some authors recommend grading LyG lesions using CD20 since this is a robust marker even in necrotic tissues (Figure 17) [38].

Additionally, diagnosing low-grade LyG in a core biopsy of a nodule does not exclude that this same nodule or other nodules may have high-grade LyG [37], since low-grade and high-grade LyG have been shown to co-exist in some cases (sampling bias issue). On small biopsies, we recommend that cases of low-grade LyG should be signed out with a comment saying that “a high-grade LyG component cannot be excluded due to sampling issues since a LyG nodule may contain admixed low-grade and high-grade areas and because the patient has multiple lung nodules”. If a high-grade LyG is sampled, then this comment is not needed, but if there is extensive necrosis, recommendation for an excisional biopsy should be done since there may not be sufficient material to establish a definitive diagnosis. Ultimately, the diagnosis made in a small biopsy should trigger the clinical team to resect one (or more) of these lesions via VATS or an open lung biopsy to have a more definitive diagnosis and avoid missing high-grade LyG or other diagnoses, such as those listed below.

*Differential diagnosis of low-grade LyG.* It includes granulomatosis with polyangiitis and polymorphic lymphoproliferative disorders with scattered large cells, such as classic Hodgkin lymphoma, T-cell/histiocyte-rich LBCL, peripheral T-cell lymphoma, not otherwise specified (PTCL-NOS), NK/T-cell lymphoma with polymorphic morphology, and EBV+ polymorphic post-transplant lymphoproliferative disorder (PTLD) [16]. Correlation with the clinical presentation (history of immunosuppression, status post chemotherapy, hereditary immunodeficiency, etc.), presence of bilateral lung nodules, and skin or central nervous system involvement are helpful to place LyG at the top of the differential diagnosis. IHC and ISH are required to confirm the diagnosis, which shows a heavy infiltration of CD3+ T-cells with scattered large EBER+/CD20+ B-cells. In classic Hodgkin lymphoma, the Reed–Sternberg cells are CD20−, CD30+, and CD15+/−, which contrasts with the immunophenotype described for the large cells of LyG. T-cell/histiocyte-rich LBCL has a similar immunophenotype to LyG, but this tumor is negative for EBER and contains abundant CD68+/CD163+ macrophages. Both classic Hodgkin lymphoma and T-cell/histiocyte-rich LBCL may contain non-necrotizing granulomas, a feature not seen in LyG, and may affect the lung only in advanced stages of disease. Therefore, prior clinical history is extremely helpful in this context.

PTCL-NOS is CD3+ in small and large cells, and NK/T-cell lymphoma shows angiocentricity with necrosis with lymphoma cells positive for CD3, CD8, CD56, and cytotoxic markers (TIA-1, granzyme-B, perforin). All these markers are negative in the large cells of LyG. Additionally, the necrosis of NK/T-cell lymphomas has numerous karyorrhectic debris, which contrasts with the eosinophilic necrosis seen in LyG devoid of debris.

The diagnosis of LyG should not be made in a patient with a history of prior bone marrow or solid organ transplantation, where the diagnosis of a polymorphic lesion with scattered EBV+ B-cells should be polymorphic PTLD, EBV+ [38]. Similarly, patients receiving methotrexate who develop lung lesions with “low-grade LyG features” should be diagnosed as iatrogenic immunodeficiency-associated lymphoproliferative disorder (LPD) and not LyG [41]. Most of these lesions are reversible once the drug is withdrawn.

*Differential diagnosis of high-grade LyG.* It is similar to that described for pulmonary DLBCL, and given the EBV positivity, it also includes EBV+ DLBCL (without prior history of transplant) and monomorphic PTLD/EBV+ DLBCL. Other differential diagnoses include primary or metastatic carcinoma or melanoma with extensive necrosis, PTCL-NOS and NK/T-cell lymphoma with large cell morphology, and anaplastic large cell lymphoma [16]. Recognition of angiocentricity supports LyG since this is not a feature seen in B-cell lymphomas but is common in extranodal NK/T-cell lymphoma and PTCL-NOS. Here again, correlation with the clinical presentation (history of immunosuppression, status post chemotherapy, hereditary immunodeficiency, etc.), presence of bilateral lung nodules, and skin or central nervous system involvement are helpful to place LyG at the top of the differential diagnosis. Primary lung DLBCL is negative for EBER, but if EBER is positive in a case suspected to be DLBCL, then the lymphoma most likely represents grade 3 LyG, and clinical history and imaging are required to confirm this finding or exclude other conditions (iatrogenic immunodeficiency associated LPD or PTLD). Establishing a diagnosis of EBV+ DLBCL in the lung should always be correlated with clinical and radiologic presentation. Most of these cases likely represent high-grade LyG (grade 3 LyG is morphologically indistinguishable from EBV+ LBCL) (Figure 16) unless the patient has a clinical presentation different from that of LyG, such as older age, lymph node, bone marrow, or spleen involvement without skin or brain involvement.

The diagnosis of high-grade LyG should not be made in a patient with a history of prior bone marrow or solid organ transplantation, where the diagnosis of monomorphic EBV+ B-cell lymphoma should be monomorphic PTLD/EBV+ DLBCL [38]. Similarly, patients receiving methotrexate who develop lung lesions with “high-grade LyG features” should be diagnosed as iatrogenic immunodeficiency-associated LPD and not LyG [41]. However, it is likely that these patients may require chemotherapy and not just withdrawal of the drug, as seen in cases with polymorphic features (see above differential diagnosis of low-grade LyG).

PTCL-NOS with large cells is CD3+, and NK/T-cell lymphoma shows angiocentricity with necrosis with lymphoma cells positive for CD3, CD8, CD56, and cytotoxic markers (TIA-1, granzyme-B, perforin). All these markers are negative in the large cells of LyG. Additionally, the necrosis seen in NK/T-cell lymphomas contains numerous karyorrhectic debris, which is not a feature of the eosinophilic necrosis seen in LyG, which is devoid of nuclear debris. For anaplastic large cell lymphoma, the use of CD3 and other T-cell markers, as well as CD4, CD8, ALK, and CD30, is helpful for diagnosis. This tumor is a CD30+ T-cell lymphoma that is usually positive for CD2 and CD4 with variable expression of CD5, CD7, and CD8 but it is negative for EBER. About half of the cases are ALK+.

Primary lung or metastatic carcinoma, melanoma, or small cell carcinoma are readily excluded from LyG by confirming the tumor cell lineage using IHC, such as pan-cytokeratin and S100, and then additional markers can be performed to further confirm if the tumor is primary or metastatic.

Lesions that are entirely necrotic are indistinguishable from a pulmonary infarct, *Pseudomonas* or *Klebsiella* pneumonia, non-infectious necrotizing pneumonias, a necrotic metastasis, or granulomatosis with polyangiitis with extensive necrosis (Figure 17). Special stains for bacteria, fungi, and acid-fast bacilli as well as correlation with microbiology studies are required to exclude infection. In granulomatosis with polyangiitis with extensive necrosis there are abundant karyorrhectic debris and neutrophils, which contrast with the “clean” coagulative/eosinophilic necrosis of LyG. Moreover, granulomatosis with polyangiitis shows multinucleated giant cells, neutrophils, and capillaritis, and it lacks the large, atypical EBER+ B-cells seen in LyG. Adding pan-cytokeratin, CD3, and CD20 IHC in cases with extensive necrosis is helpful to attempt to rule out a necrotic metastasis (highlighting the “ghost” tumor cells with keratin) or a necrotic LBCL (highlighting “ghost” tumor cells positive for CD20 and negative for CD3) (Figure 17). The latter, in the right clinical and radiologic context, could suggest LyG but could not exclude DLBCL. If no specific findings are obtained, the recommendation should be to request an open lung biopsy, VATS biopsy, or a larger resection with additional material submitted for microbiology studies. If obtained, these lesions require extensive sampling to attempt to identify potential areas of LyG not present in the biopsy.

We discourage submitting material for flow cytometry and/or clonality studies if the clinical impression and diagnostic consideration is LyG. Although *IGH* gene rearrangements have been detected in grade 2 and grade 3 LyG with more frequency than in grade 1 LyG (50% in grade 2, 70% in grade 3, and <10% in grade 1) [37], the presence of necrosis may limit the viability of a sample and correct interpretation. Flow cytometry may not detect the already few large B-cells present in a cellular sample in low-grade LyG or may show non-specific staining due to extensive necrosis in high-grade LyG. Morphology, IHC, and ISH remain the gold standard for diagnosis.

## 6. Conclusions

Given the current use of less invasive methods for sampling lung lesions, the diagnosis of pulmonary neoplasms, including pulmonary lymphomas, has become more challenging. Thus, pathologists are now required to know more about the approach to these lesions in smaller samples, such as a core needle biopsy, a transbronchial biopsy, or a fine-needle aspiration. Because of its rarity, a pulmonary lymphoma may not be originally submitted for a “lymphoma protocol”; therefore, material submitted for flow cytometry or cytogenetics is usually not obtained. For this reason, we provide herein recommendations for the diagnosis of some of the most common pulmonary lymphomas in small biopsies. It should be emphasized that clinical and imaging correlation is mandatory to avoid making a wrong diagnosis, missing a diagnosis, or overcalling a reactive process lymphoma. The best recommendation a pathologist can make after reviewing a small lung biopsy not conclusive for lymphoma diagnosis is to recommend additional sampling (open lung biopsy, VATS biopsy, wedge resection) with material submitted for flow cytometry, cytogenetics and/or molecular studies to be able to properly classify a pulmonary lymphoid process. The recommendation of additional sampling should also be discussed with the clinical team when a diagnosis of lymphoma has been made but additional prognostic or predictive markers cannot be performed and are needed for further therapeutic decisions, as detailed for each entity in this manuscript.

## Figures and Tables

**Figure 1 diagnostics-13-03321-f001:**
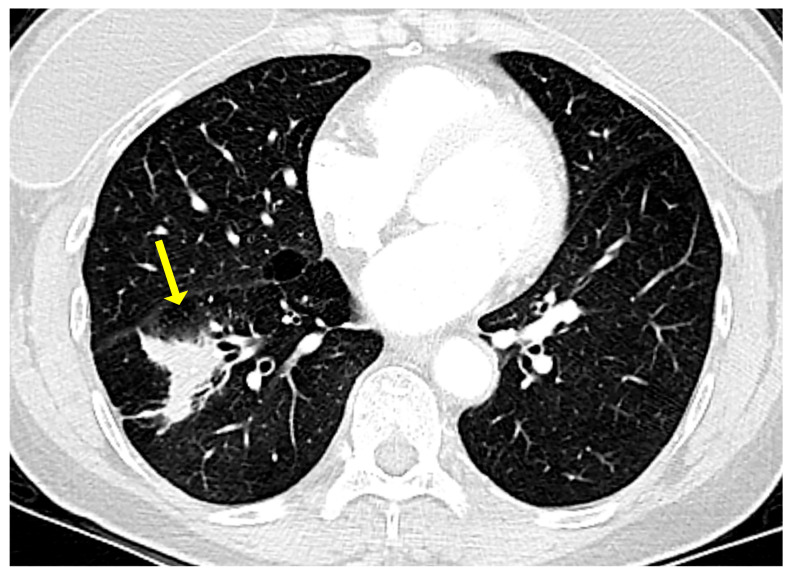
Computed tomography in a case of pulmonary marginal zone lymphoma of mucosa-associated lymphoid tissue (MALT lymphoma). There is an ill-defined lesion in the right lower lobe originally thought to represent pneumonia (arrow). This lesion did not resolve with antibiotic therapy. Note the air bronchograms towards the medial aspect of the lesion.

**Figure 2 diagnostics-13-03321-f002:**
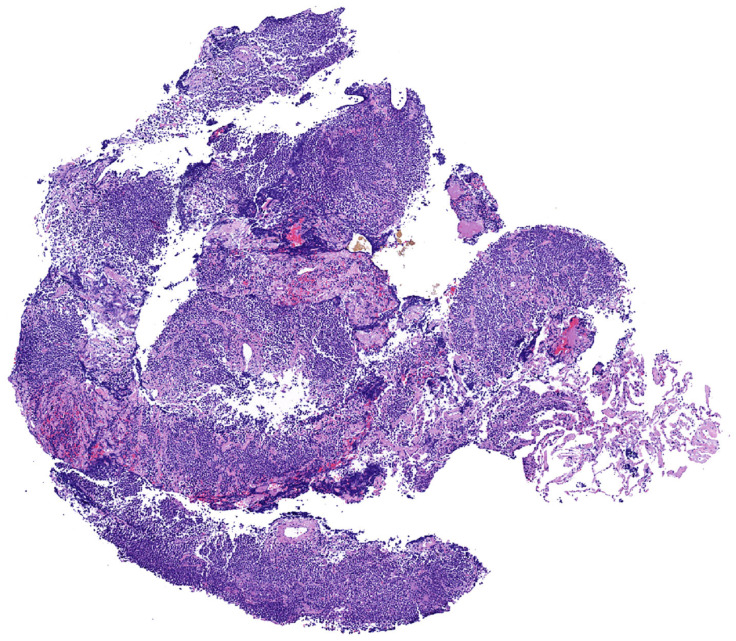
Pulmonary MALT lymphoma; core biopsy. There is effacement of the lung architecture by small cell lymphoma with associated areas of fibrosis.

**Figure 3 diagnostics-13-03321-f003:**
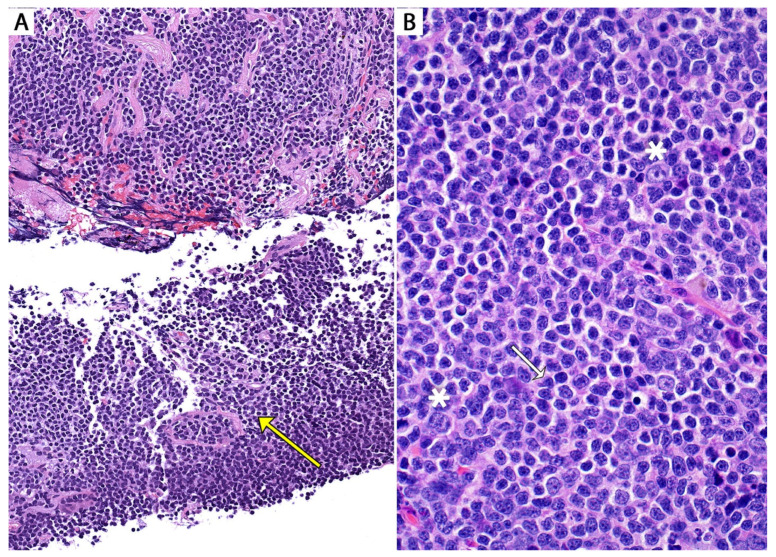
Pulmonary MALT lymphoma; core biopsy. (**A**) Monotonous infiltrate of small lymphocytes replacing the lung parenchyma. Two lymphoepithelial lesions are seen. Note the profile of the residual basement membranes (yellow arrow). (**B**) Cytologic features of MALT lymphoma. Numerous small lymphocytes with clear cytoplasm (monocytoid cells), centrocyte-like cells (white arrow), and a few scattered immunoblast-like cells (white asterisks).

**Figure 4 diagnostics-13-03321-f004:**
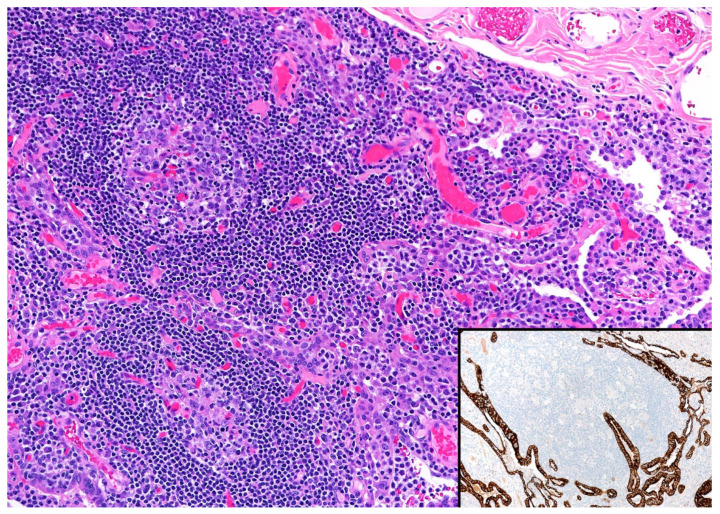
Pulmonary nodular lymphoid hyperplasia. The alveolar septa are expanded by reactive lymphoid follicles with prominent germinal centers and a few interfollicular plasma cells. Inset, pan-cytokeratin highlights displaced epithelium by the reactive lymphoid process, but no lymphoepithelial lesions.

**Figure 5 diagnostics-13-03321-f005:**
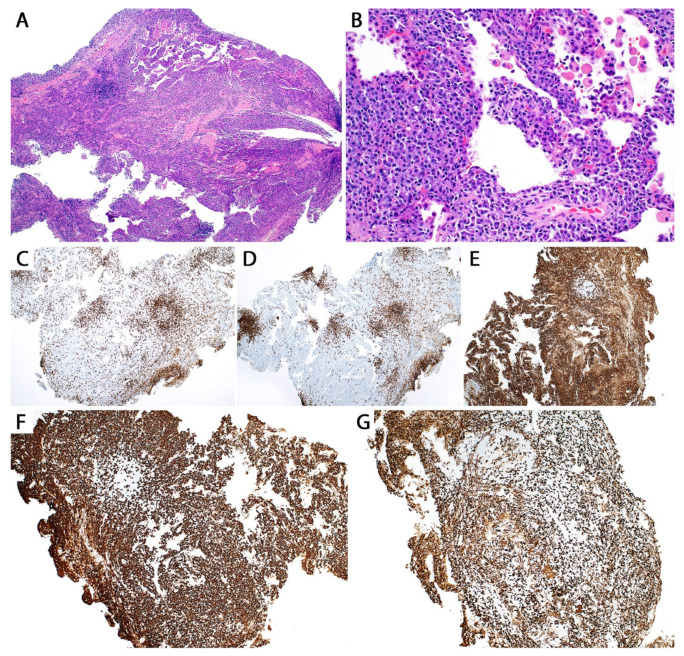
(**A**) Brisk pulmonary lymphoplasmacytic infiltrate in a patient with Sjögren syndrome; transbronchial biopsy. Imaging showed ground glass opacities. (**B**) Higher magnification shows alveolar septal expansion due to a plasma cell-rich infiltrate. (**C**) CD3 and (**D**) CD20 show a few small lymphoid follicles composed of a mixture of T-cells and B-cells, respectively. (**E**) CD138 highlights numerous plasma cells, polytypic by (**F**) kappa and (**G**) lambda. *IGH* clonality studies were negative. The differential diagnosis included involvement by the patient’s known connective tissue disease, lymphoid interstitial pneumonia, and less likely MALT lymphoma with plasmacytic differentiation, but the latter could not be entirely excluded. With these results, the patient underwent wedge resection that showed identical features. Flow cytometry was negative for monotypic B-cells/plasma cells, and repeated *IGH* clonality studies were negative. The final diagnosis was extensive lung involvement by the patient’s known Sjögren syndrome. Close follow up was recommended since these patients may progress to MALT lymphoma without treatment. The patient showed significant improvement with steroids.

**Figure 6 diagnostics-13-03321-f006:**
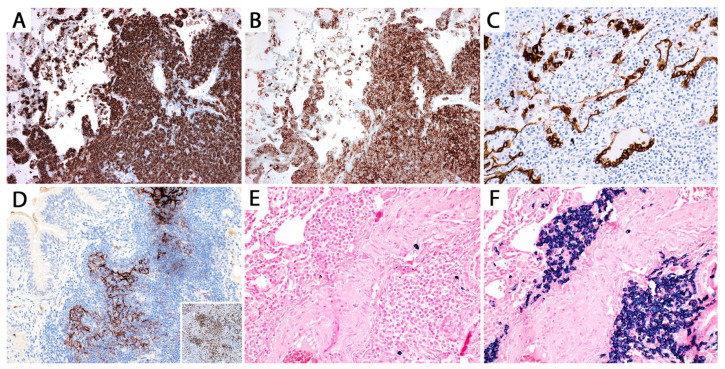
Pulmonary MALT lymphoma, immunohistochemistry. (**A**) CD20. (**B**) CD43 is co-expressed in B-cells in 40–50% of cases. (**C**) Pan-cytokeratin highlights lymphoepithelial lesions. (**D**) CD23 decorates distorted follicular dendritic cell meshworks due to follicular colonization. Inset: bcl-6 also highlights the distorted germinal centers in these same areas. Clusters of plasma cells in this same process are negative for (**E**) kappa and positive for (**F**) lambda by in situ hybridization.

**Figure 7 diagnostics-13-03321-f007:**
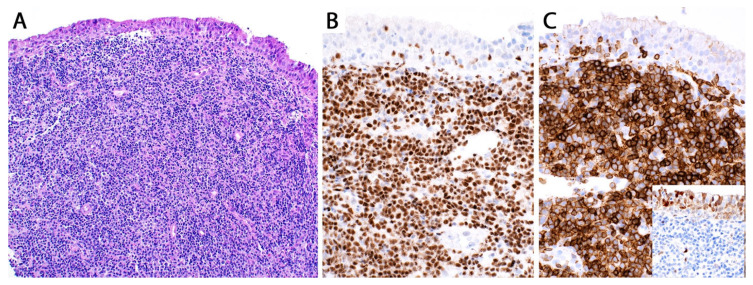
Differential diagnosis of pulmonary MALT lymphoma. (**A**) Small cell lymphoma with occasional “monocytoid cells” replacing the bronchial submucosa. The lymphoma cells are positive for (**B**) PAX5 and (**C**) CD5, and negative for cyclin D1 ((**C**), inset). The patient had a diagnosis of refractory chronic lymphocytic leukemia/small lymphocytic lymphoma and presented with severe cough and multiple endobronchial lesions.

**Figure 8 diagnostics-13-03321-f008:**
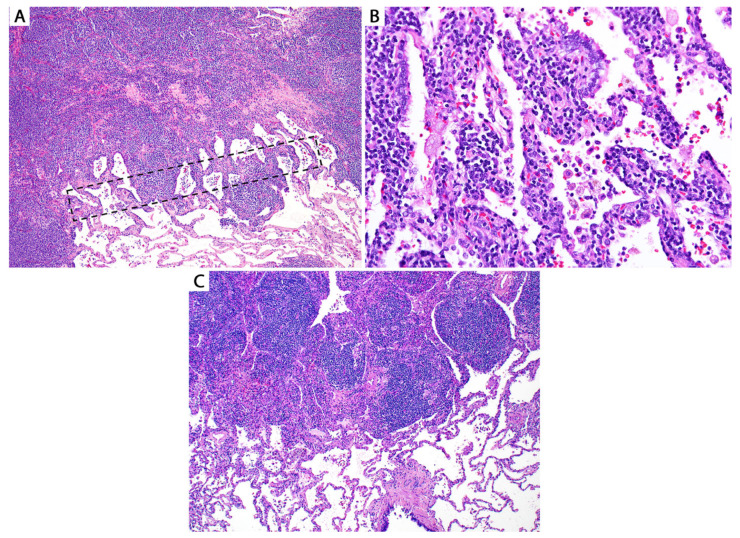
(**A**) Edge of a pulmonary MALT lymphoma. If a core needle biopsy passes through the delineated area, the morphologic features will be indistinguishable from (**B**) lymphoid interstitial pneumonia (LIP). Imaging correlation is mandatory in this case since MALT lymphoma usually presents as a mass, whereas LIP presents as ground glass opacities. (**C**) In contrast to MALT lymphoma, the edge of nodular lymphoid hyperplasia is sharply demarcated from the lung parenchyma.

**Figure 9 diagnostics-13-03321-f009:**
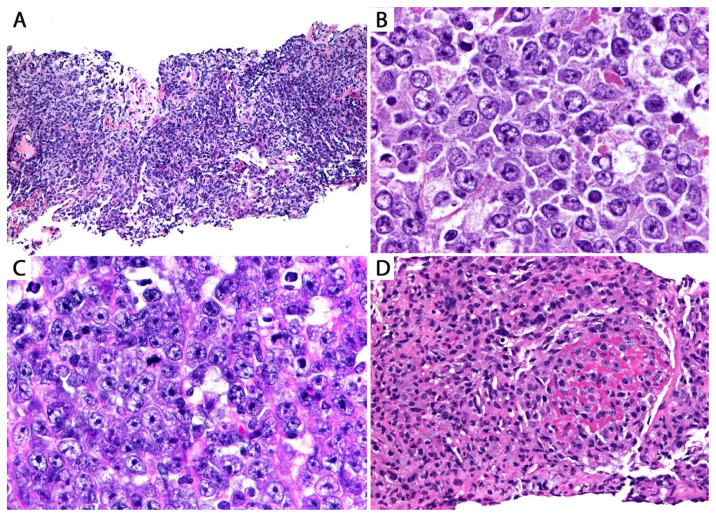
(**A**) Pulmonary diffuse large B-cell lymphoma (DLBCL), core biopsy. There is effacement of the lung architecture by large lymphoma cells. Some cells show an elongated nucleus. (**B**) Predominantly centroblastic morphology. (**C**) Predominantly immunoblastic morphology (see text). (**D**) Tumoral pneumonia, seen as intra-alveolar lymphoma cells admixed with fibrin and alveolar macrophages.

**Figure 10 diagnostics-13-03321-f010:**
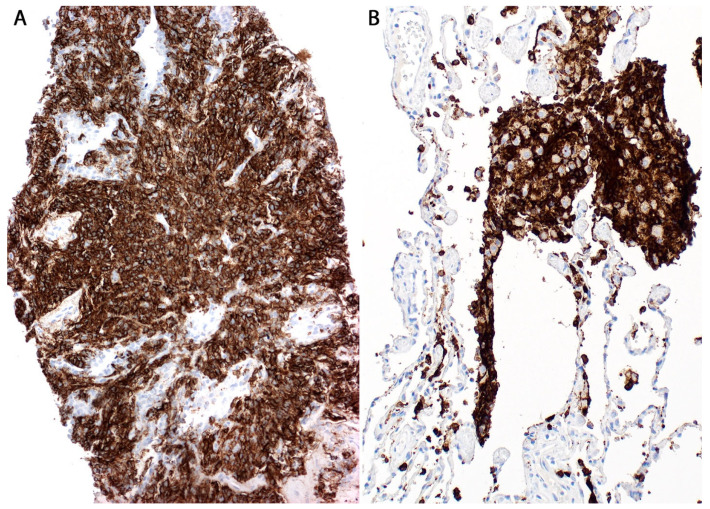
Pulmonary DLBCL, core biopsy. (**A**) CD20. (**B**) Tumoral pneumonia, CD79a immunostain. The partially necrotic lymphoma cells are positive, whereas the admixed alveolar macrophages are negative.

**Figure 11 diagnostics-13-03321-f011:**
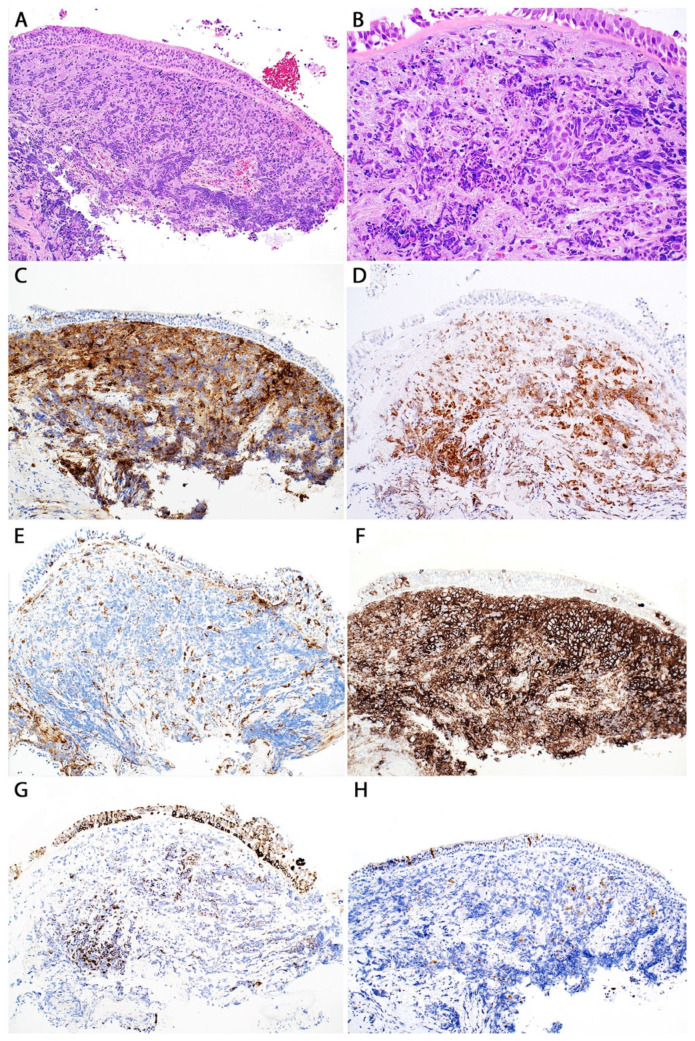
Differential diagnosis of pulmonary DLBCL. Transbronchial biopsy of a 70-year-old man with a history of heavy smoking with a large hilar mass who also had a history of CD10+ DLBCL in remission. (**A**,**B**) The morphology suggests small cell carcinoma, but DLBCL could not be excluded given the prior history and the marked crush artefact. The tumor cells are positive for (**C**) CD10 and (**D**) PAX5, potentially misleading to a diagnosis of relapsed CD10+ DLBCL. CD20 was negative (not shown) but the patient had received rituximab in the past. Additional immunohistochemical stains show that the tumor cells are negative for (**E**) CD45, positive for (**F**) CD56, and focally positive for (**G**) pan-cytokeratin and (**H**) synaptophysin, confirming small cell carcinoma.

**Figure 12 diagnostics-13-03321-f012:**
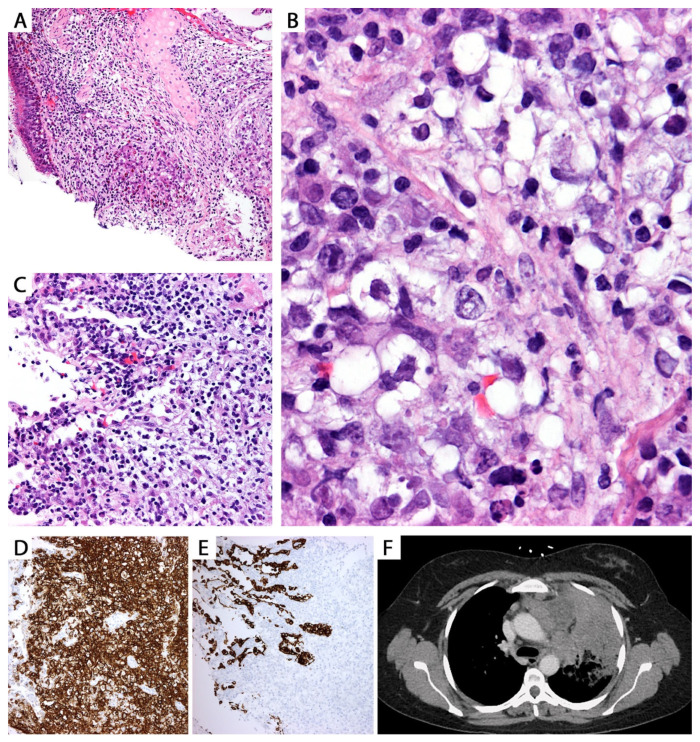
Differential diagnosis of pulmonary DLBCL. (**A**–**C**) The transbronchial biopsy shows sheets of malignant cells with an oval nucleus and clear cytoplasm infiltrating the bronchus and the adjacent lung parenchyma. The tumor cells are positive for (**D**) CD20 and negative for (**E**) cytokeratin 7. A diagnosis of pulmonary DLBCL can be entertained; however, (**F**) the clinico-radiologic presentation is that of a young woman with a prevascular/anterior mediastinal mass widely involving the lung and visceral mediastinum. This supports the diagnosis of primary mediastinal (thymic) LBCL with lung involvement rather than pulmonary DLBCL. Additional immunostains favored the former (positive MUM1 and CD23 and weak CD30) (not shown).

**Figure 13 diagnostics-13-03321-f013:**
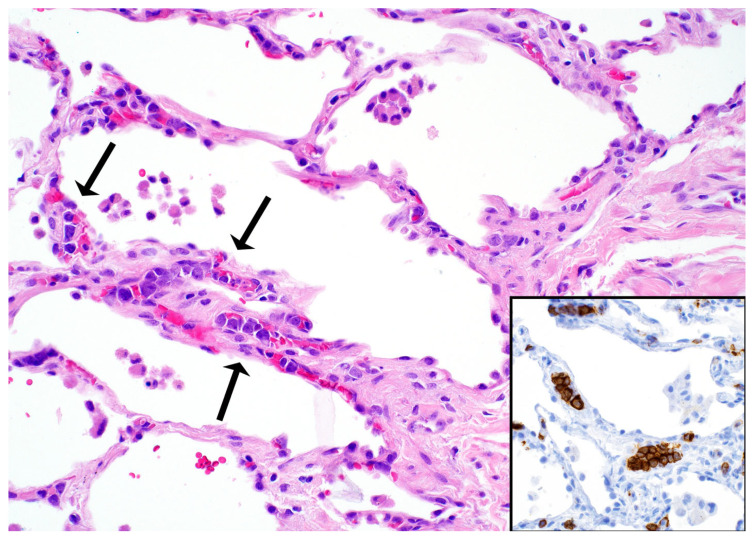
Intravascular large B-cell lymphoma (arrows). Inset: CD20 highlights the lymphoma cells within capillaries.

**Figure 14 diagnostics-13-03321-f014:**
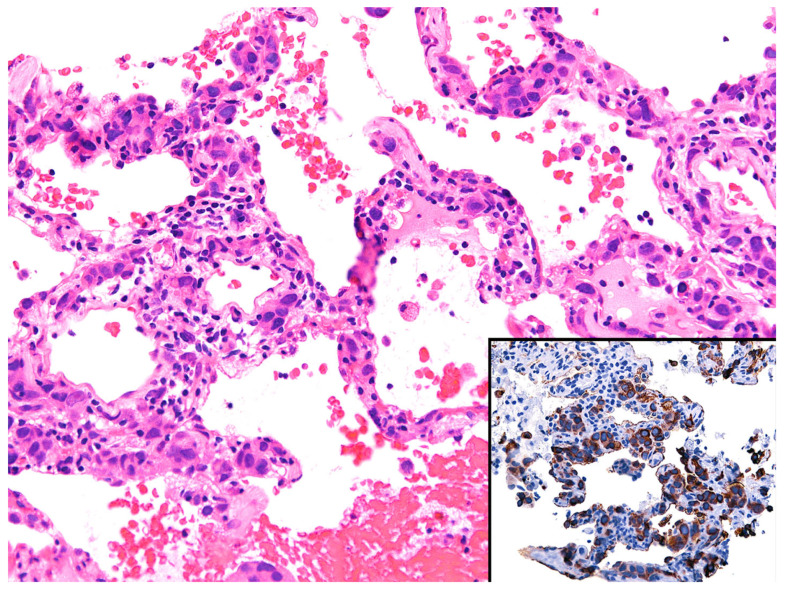
Differential diagnosis of intravascular large B-cell lymphoma. Lymphangitic carcinoma in a patient with a history of invasive ductal carcinoma. Inset: CK7 highlights numerous intravascular carcinoma cells, which were also positive for GATA3 (not shown).

**Figure 15 diagnostics-13-03321-f015:**
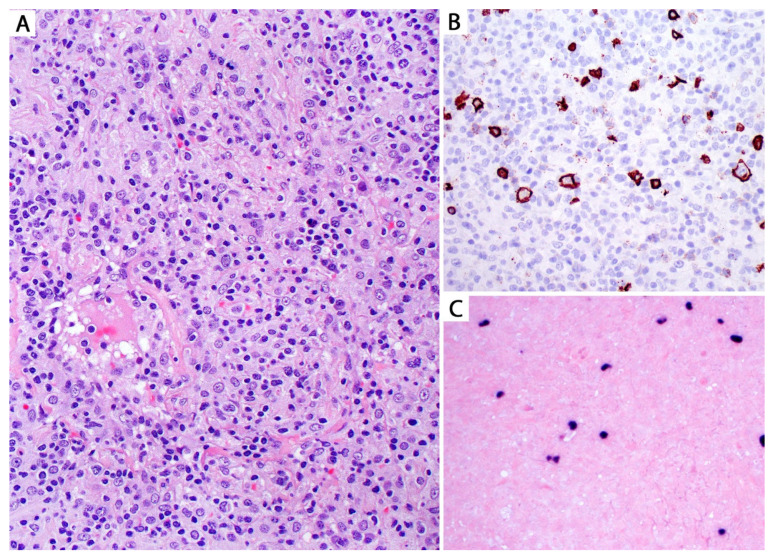
(**A**) Low-grade (grade 1–2) lymphomatoid granulomatosis (LyG). Polymorphic infiltrate composed of small lymphocytes, macrophages, plasma cells, and a few scattered large atypical cells without necrosis (grade 1). The large atypical cells are positive for (**B**) CD20 and (**C**) EBER by in situ hybridization. Based on the number of these cells per high power field and the presence or absence of necrosis, a case may be graded 1 or 2 (see text). On a biopsy, a diagnosis of low-grade LyG can be rendered with a comment saying that high-grade LyG cannot be excluded since LyG has been shown to feature low- and high-grade areas in a same nodule.

**Figure 16 diagnostics-13-03321-f016:**
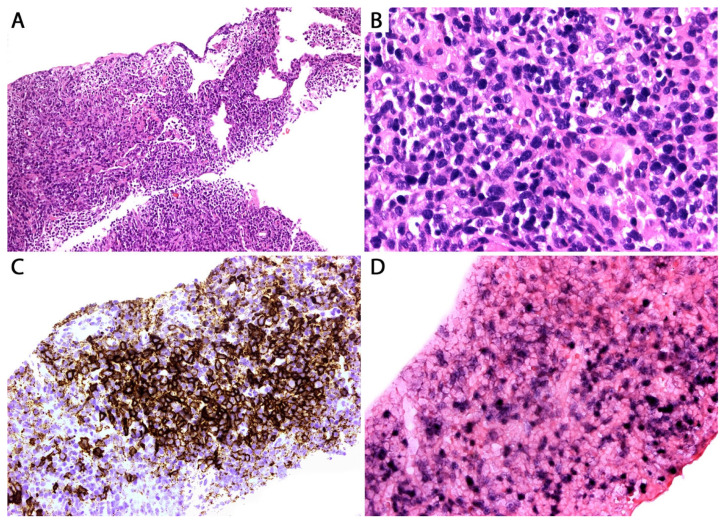
(**A**) High-grade (grade 3) LyG; core biopsy. (**B**) The infiltrate consists of sheets of intermediate to large lymphoid cells effacing the lung parenchyma. Other areas showed necrosis, but vasculitis was not seen (not uncommon in a biopsy). The lymphoma cells are positive for (**C**) CD20 and for (**D**) EBER by in situ hybridization. The patient was immunosuppressed and developed multiple lung nodules without lymphadenopathy or bone marrow involvement. A diagnosis of high-grade LyG was favored. This morphology is indistinguishable from EBV+ LBCL. Clinical and imaging correlation are the only way to support LyG (see text).

**Figure 17 diagnostics-13-03321-f017:**
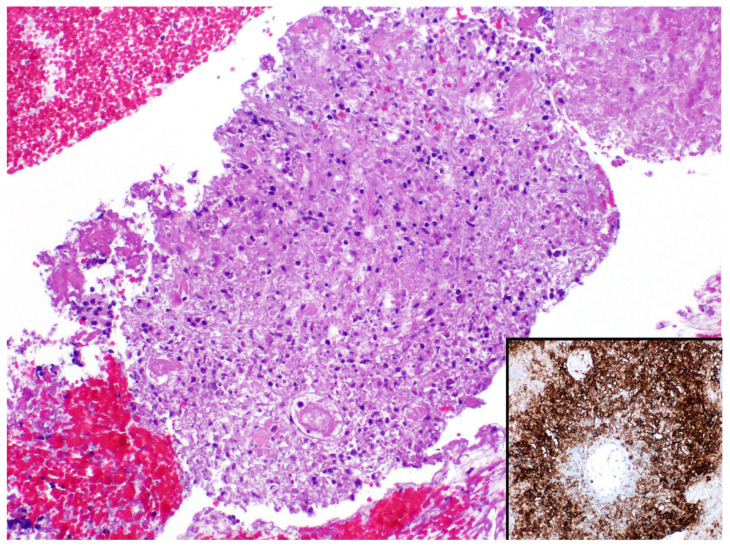
High-grade (grade 3) LyG with extensive necrosis; core biopsy. The differential diagnosis in this setting is wide, including a necrotizing infection, a lung infarct, necrotic metastases, necrotic lymphoma, granulomatosis with polyangiitis, etc. Inset: CD20 confirms the presence of CD20+ “ghost” tumor cells around a blood vessel. EBER in situ hybridization was weak and patchy positive. GMS and AFB were negative for microorganisms. The patient had multiple lung nodules as well as brain and skin nodules. A biopsied skin nodule confirmed high-grade LyG. The lung biopsy was signed out as necrotic B-cell lymphoma, consistent with LyG given the patient’s history.

**Table 1 diagnostics-13-03321-t001:** Differential diagnosis of pulmonary B-cell lymphomas on small biopsies.

Lymphoma Subtype	Differential Diagnosis
Marginal zone lymphoma of mucosa-associated lymphoid tissue (MALT lymphoma)	*Non-neoplastic:* − Non-specific chronic lymphoplasmacytic inflammation (if mass-forming). − Nodular lymphoid hyperplasia. − IgG4-related lung disease. − Lymphocytic interstitial pneumonia (usually not a mass, but may be identical on biopsy to sampling the edge of a MALT lymphoma). *Neoplastic:* − Chronic lymphocytic leukemia/small lymphocytic lymphoma. − Mantle cell lymphoma. − Follicular lymphoma. − Lymphoplasmacytic lymphoma. − Solitary plasmacytoma or extramedullary plasma cell myeloma (particularly MALT lymphoma with extensive plasmacytic differentiation).
Diffuse large B-cell lymphoma (DLBCL)	*Hematolymphoid:* − Primary mediastinal (thymic) LBCL extending to lungs. − Burkitt lymphoma (exceedingly rare in the lung). − Anaplastic large cell lymphoma. − Grade 3 (high grade) LyG. − Other hematopoietic tumors: Myeloid sarcoma. *Non-hematolymphoid:* − Poorly-differentiated carcinoma, including small cell carcinoma. − Metastatic melanoma. − Small blue round cell tumors. − Sarcoma or sarcomatoid carcinoma (DLBCL can show spindle cell artefactual changes).
Intravascular large B-cell lymphoma (IV-LBCL)	Diagnosis may be missed due to the lack of significant pathology, or only minimal alveolar septal expansion *Hematolymphoid:* − Intravascular T-cell lymphoma (NK/T-cell). − Intravascular anaplastic large cell lymphoma. − Acute leukemia with hyperleukocytosis. *Non-hematolymphoid:* − Carcinoma or melanoma with lymphangitic spread.
Lymphomatoid granulomatosis (LyG)	Differential diagnosis depends on grading (see also text). *Low-grade LyG (grade 1-2 with no or focal necrosis):* − Granulomatosis with polyangiitis. − Classic Hodgkin lymphoma. − T-cell / histiocyte-rich LBCL. − Peripheral T-cell lymphoma, not otherwise specified, with polymorphic morphology. − NK/T-cell lymphoma, with polymorphic morphology. − Post-transplant lymphoproliferative disorder, Epstein-Barr virus (EBV)+ *(history of transplant excludes LyG).* − Iatrogenic immunodeficiency-associated lymphoproliferative disorder, EBV+ *(history of methotrexate treatment excludes LyG).* *High-grade LyG (grade 3, with necrosis):* *Hematolymphoid:* − DLBCL. − EBV+ DLBCL (histopathology identical to high-grade LyG; mandatory to know clinical history and imaging features to distinguish). − Monomorphic post-transplant lymphoproliferative disorder, EBV+ *(history of transplant excludes LyG).* − Iatrogenic immunodeficiency associated lymphoproliferative disorder, EBV+ *(history of methotrexate treatment excludes LyG).* − Peripheral T-cell lymphoma, not otherwise specified, with large cells. − NK/T-cell lymphoma, with large cells. − Anaplastic largecell lymphoma *High grade LyG (grade 3, with necrosis):* *Non-hematolymphoid:* − Poorly-differentiated carcinoma, including small cell carcinoma. − Metastatic melanoma. *Grade 2 or 3 LyG with extensive necrosis:* − Pulmonary infarct. − Necrotizing pneumonia (*Pseudomonas, Klebsiella,* fungal or mycobacterial). − Necrotic metastasis. − Granulomatosis with polyangiitis with extensive necrosis (necrosis shows abundant karyorrhectic debris as compared to LyG).

## Data Availability

Not applicable.

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
