# Peer review of "Diagnostic Approach to Pulmonary B-Cell Lymphomas in Small Biopsies, with Practical Recommendations to Avoid Misinterpretation"

_diagnostics, 2023, doi:10.3390/diagnostics13213321_

Round 1

Reviewer 1 Report

It has been my experience that the diagnosis of lymphoma lung lesions is difficult to make pathologically, because only small and distorted tissue fragments can be obtained. This paper focuses on this problem and provides a comprehensive and detailed review of the pathologic diagnosis of lymphoma in the lung for the pathologist, which I believe will be useful to readers.

Table 1 is illegible and should be corrected. It should be left-aligned.

The authors mention the importance of the pathologist suggesting further tissue biopsy to the clinician. It would be helpful to add a scheme or other information on when a pathologist should take such action.

Author Response

Reviewer 1

It has been my experience that the diagnosis of lymphoma lung lesions is difficult to make pathologically, because only small and distorted tissue fragments can be obtained. This paper focuses on this problem and provides a comprehensive and detailed review of the pathologic diagnosis of lymphoma in the lung for the pathologist, which I believe will be useful to readers.

The authors thank the reviewer’s nice comments about our manuscript.

Table 1 is illegible and should be corrected. It should be left-aligned.

We agree with the reviewer. The table was formatted this way by the journal and not by us. We will let the journal and the editor of the Special Issue to left-aligned all bullets on Table 1.

The authors mention the importance of the pathologist suggesting further tissue biopsy to the clinician. It would be helpful to add a scheme or other information on when a pathologist should take such action.

We thank the reviewer for this comment. We have added at the end of the conclusion the following sentence (underlined):

The best recommendation a pathologist can make after reviewing a small lung biopsy not conclusive for lymphoma diagnosis is to recommend additional sampling (open lung biopsy, VATS biopsy, cryobiopsy, wedge resection) with material submitted for flow cytometry, cytogenetics and/or molecular studies to be able to properly classify a pulmonary lymphoid process. The recommendation of additional sampling should be also discussed with the clinical team when a diagnosis of lymphoma has been made but additional prognostic or predictive markers cannot be performed and are needed for further therapeutic decisions, as detailed for each entity in this manuscript.

We thank the reviewer for the comments and helpful suggestions. We have addressed the reviewer’s comments, and we therefore hope the revised manuscript is acceptable for publication.

Sincerely,

The authors

Reviewer 2 Report

please see file enclosed

Author Response

Reviewer 2

The authors report on the diagnostics (including differential diagnoses) of B cell lymphomas of the lungs. The manuscript is well written and the authors report a good overview of these rare diseases. I have some comments:

The manuscript is very long, some parts might be shortened.

The authors thank the reviewer’s nice comments. We believe the length of the manuscript is adequate. Our original idea was to include several other lymphomas which could certainly be included (plasma cell neoplasms, T-cell lymphomas, acute leukemia, etc), but this would have made the review quite more extensive.

I table summarizing the most important lymphoma subtypes including morphological characteristics, differential diagnosis and recommended sample types (e.g. fine needle, VATS etc.) might be helpful (on contrary, some of the many figures might be removed to keep the paper more concise)

Thank you for the suggestion. The recommended biopsy approach varies according to the location and features of the lung lesion and not by the lymphoma subtype. Therefore, the sample types cannot be associated with a specific lymphoma diagnosis. Table 1 lists the differential diagnoses for each entity. We did not add another table listing morphologic findings for each lymphoma to avoid repetition in the text and again in a table. Regarding the figures, we believe that the figures add more value to the manuscript since they clearly illustrate examples of the kind of cases that give trouble to pathologists for diagnosis.

Plasmocytoma should be classified according to IMWG criteria (e.g. solitary plasmocytoma, extramedullary multiple myeloma etc.)

We have included these classifiers for plasmacytoma in the text and Table 1 in the appropriate sections.

Table 1.: I assume that myeloid sarcoma does not belong to hematolymphoid diseases?

Thank you for noticing this semantic misconception. We have modified this sentence on Table 1 to the following: “ Other hematopoietic tumors: Myeloid sarcoma ”.

Figure 1. An arrow showing the main abnormality might be helpful.

We have added and arrow pointing to the lesion on Figure 1. This was also added to the figure legend.

Page 6: `plasma cells are always polytypic´. The authors refer here to immunophenotypic characteristics or morphology?

We refer to the immunophenotypic features. We have modified this sentence to: “plasma cells are always polytypic by IHC and/or ISH”.

Please check that all abbreviations such as RLD are described.

We have checked that all abbreviations, including RLD (related lung disease), are appropriately listed and described throughout the text.

Page 8. `IgG4:IgG ratio´ refers to serum IgG or plasma cells?

This refers to plasma cells, not serum, since the sentence is referring to the Deshpande criteria for IgG4-disease which is used in tissues.

I would suggest to write `patients´ instead of `cases´.

We used the term “cases” rather than “patients” since most of the information of the manuscript refers to pathologic entities rather than individuals.

Page 10: I suggest to substitute the expression `fortunately´

We have changed the word “fortunately” for “Usually”.

We thank the reviewer for the comments and helpful suggestions. We have addressed the reviewer’s comments, and we therefore hope the revised manuscript is acceptable for publication.

Sincerely,

The authors

Reviewer 3 Report

I read the article entitled "Diagnostic approach to pulmonary B-cell lymphomas in small biopsies, with practical recommendations to avoid misinterpretation" and I would like to congratulate the authors on an exhaustive narrative review of lung lymphomas and their histopathological particularities.

It is a well-written paper, but it lacks a "Discussion" chapter which I consider important and necessary given some issues that have to be addressed and improved.

However, in order to be published I believe that the authors should :

- comment on what biopsy approach is more suitable in order to ensure the correct diagnosis of lung lymphomas

-mention the need or the challenge of having ROSE in the endoscopy department (Rapid On Site Examination)

- an honest comparison between EBUS, transbronchial biopsy and criobiopsy

- costs of diagnosing pulmonary lymphomas in classical surgery vs minimal 

- a correlation of the rate of diagnostic success and radiological findings so the healthcare professionals cand learn (if there is any) that some particular radiological presentations can be easily diagnosed by minimal interventions such EBUS.

Thank you.

Please add and extensive "Discussion" chapter and address the issues above but some more that you will think of.

Thank you. 

Author Response

Reviewer 3

I read the article entitled "Diagnostic approach to pulmonary B-cell lymphomas in small biopsies, with practical recommendations to avoid misinterpretation" and I would like to congratulate the authors on an exhaustive narrative review of lung lymphomas and their histopathological particularities.

The authors thank the reviewer’s nice comments about our manuscript.

It is a well-written paper, but it lacks a "Discussion" chapter which I consider important and necessary given some issues that have to be addressed and improved.

However, in order to be published I believe that the authors should :

- comment on what biopsy approach is more suitable in order to ensure the correct diagnosis of lung lymphomas

We thank the reviewer for the comments. The recommended biopsy approach varies according to the location and features of the lung lesion and not by the lymphoma subtype. Therefore, the biopsy approach falls in the hands of the clinical or interventional radiology teams, and it is out of the scope of this review that is directed to pathologists.

-mention the need or the challenge of having ROSE in the endoscopy department (Rapid On Site Examination)

The purpose of this manuscript is to give practical diagnostic recommendations to pathologists, and not to address the issues occurring at endoscopy, such as ROSE. This may be a suitable topic to discuss in a manuscript addressing different types of procedural methods in the setting of lung lymphoproliferative disorders.

- an honest comparison between EBUS, transbronchial biopsy and criobiopsy

The purpose of this manuscript is diagnosis of pulmonary lymphomas in small biopsies rather than discussion or comparison of methods to obtain these biopsies. Therefore, this topic is out of the scope of this review. However, as requested by the reviewer, we searched the literature about some of these methods and have included information about the use of cryobiopsies for lung lymphomas. We have included a new sentence in the Introduction that reads: “A retrospective study from Italy has shown that a transbronchial cryobiopsy may be also an effective tool in the diagnosis of pulmonary lymphomas (new reference: Bianchi et al ERJ Open Res 2020; 6: 00260-2019).

- costs of diagnosing pulmonary lymphomas in classical surgery vs minimal 

This topic is out of the scope of this review. This may be a suitable topic to discuss in a manuscript addressing different types of procedural methods in the setting of lung lymphoproliferative disorders.

- a correlation of the rate of diagnostic success and radiological findings so the healthcare professionals cand learn (if there is any) that some particular radiological presentations can be easily diagnosed by minimal interventions such EBUS.

This topic is out of the scope of this review. This may be a suitable topic to discuss in a manuscript addressing different types of procedural methods in the setting of lung lymphoproliferative disorders. In addition to that, the issue with the diagnosis of pulmonary lymphomas is that they mimic more common entities clinically and radiologically, and the diagnosis is usually not suspected clinically.

Thank you.

Please add and extensive "Discussion" chapter and address the issues above but some more that you will think of.

Each described entity already contains a section of differential diagnosis where the main issues for diagnosis in small biopsies are discussed. Therefore, we don’t believe that an extensive discussion is needed and in fact will make the manuscript too long. Additionally, the main purpose of this manuscript is to give practical recommendations to pathologists for the diagnosis of pulmonary lymphomas in small biopsies rather than to discuss or compare methods to obtain these biopsies.

We thank the reviewer for the comments and helpful suggestions. We have addressed the reviewer’s comments, and we therefore hope the revised manuscript is acceptable for publication.

Sincerely,

The authors

Round 2

Reviewer 2 Report

thanks, I do not have further comments

Reviewer 3 Report

The authors addressed the issues suggested and I believe the article should be published in the present form